# *AdaBlock-dLLM*: Semantic-Aware Diffusion LLM Inference via Adaptive Block Size

**Guanxi Lu**[1]   **Hao (Mark) Chen**[1]   **Yuto Karashima**[2]
**Zhican Wang**[1]   **Daichi Fujiki**[2]   **Hongxiang Fan**[1]
[1]Imperial College London, UK   [2]Institute of Science Tokyo, Japan
{*guanxi.lu22, hao.chen20*}*@imperial.ac.uk, karashima.yuto@artic.iir.isct.ac.jp*
*wangzhican1999@gmail.com, dfujiki@artic.iir.isct.ac.jp, hongxiang.fan@imperial.ac.uk*

## Abstract

Diffusion-based large language models (dLLMs) are gaining attention for their inherent capacity for parallel decoding, offering a compelling alternative to autoregressive LLMs. Among various decoding strategies, block-wise semi-autoregressive (semi-AR) approaches are widely adopted due to their support for KV caching and their favorable accuracy–speed trade-off. However, this paper identifies two fundamental limitations in the conventional semi-AR decoding approach that applies a fixed block size: *i)* late decoding overhead, where the unmasking of high-confidence tokens outside the current block is unnecessarily delayed, and *ii)* premature decoding error, where low-confidence tokens inside the current block are committed too early, leading to incorrect tokens. This paper presents the first systematic investigation challenging the fixed block size setting in semi-AR decoding. Through a statistical analysis of confidence dynamics during the denoising process, we identify a volatility band (VB) region during dLLM decoding, which encodes local semantic structure and can be used to guide adaptive block sizing. Leveraging these insights, we introduce *AdaBlock-dLLM*, a training-free, plug-and-play scheduler that adaptively aligns block boundaries with semantic steps by adjusting block size during runtime. Extensive experiments across diverse benchmarks show that *AdaBlock-dLLM* achieves up to $5.3\%$ accuracy improvement under the same throughput budget. Beyond inference-time optimization, we hope our semantics-aware adaptive scheduling approach and confidence-based analysis will inspire future training strategies for dLLMs. Our code is available at https://github.com/lgxi24/AdaBlock-dLLM.

## 1 Introduction

Diffusion-based large language models (dLLMs) have recently emerged as a promising alternative to autoregressive models, offering parallel decoding, improved controllability, and greater data efficiency in low-resource settings (Zhang et al., 2025; Prabhudesai et al., 2025). Open-source dLLMs such as LLaDA (Nie et al., 2025; Zhu et al., 2025) and Dream (Ye et al., 2025) have demonstrated comparable performance to autoregressive models of similar scale. Notably, in structured generation tasks such as coding, proprietary models including Seed Diffusion (Song et al., 2025b) and Gemini Diffusion (Gemini Diffusion, 2025) have achieved throughput exceeding $1,400$ tokens per second. These advances highlight the potential of dLLMs to deliver efficient inference while maintaining competitive algorithmic performance.

Recent works have widely adopted a semi-autoregressive (semi-AR) decoding paradigm that combines block-wise KV caching (Wu et al., 2025; Chen et al., 2025; Song et al., 2025a) and confidence-based dynamic sampling (Wang et al., 2025d; Wu et al., 2025; Wei et al., 2025) to improve inference efficiency. However, semi-AR decoding enforces block-level causality: the current block must be finalized before decoding the next block. We identify two **fundamental issues** introduced by conventional semi-AR decoding with a fixed block size: *i)* **Late Decoding Overhead.** As shown in the upper-left of Figure 1, semi-AR decoding delays the unmasking of high-confidence tokens outside the current block. For instance, the second and third blocks are decoded in separate iterations, incurring unnecessary computational overhead to generate a simple complete sentence. *ii)* **Premature Decoding Error.** As shown in the lower-left of Figure 1, the autoregressiveness across blocks forces

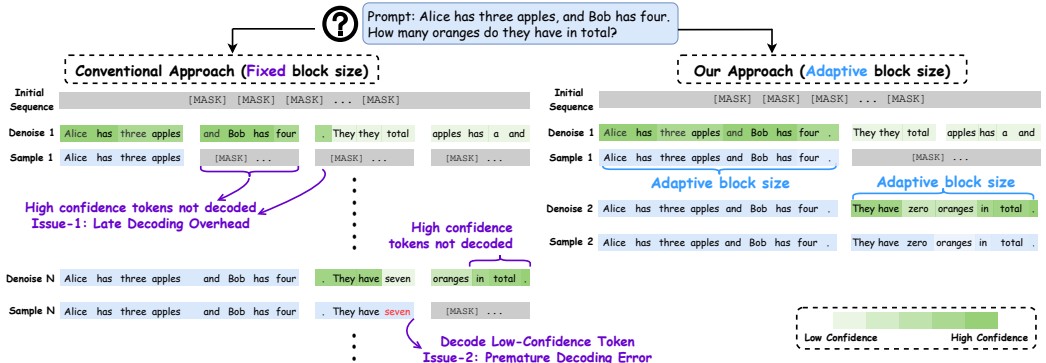

Figure 1: Illustrative examples of two fundamental issues (left) and how *AdaBlock-dLLM* addresses them (right). Appendix A.1 presents a case study from a real inference scenario.

early commitment to low-confidence tokens within each block, and this suboptimal sampling often yields incorrect token predictions (Figure 5), particularly in reasoning tasks.

To address these limitations, we begin by investigating how confidence scores evolve across decoding steps. Our statistical analysis identifies a volatility band (VB), a region in which confidence fluctuates markedly over time. VB regions encode local semantic structure, which can be exploited to dynamically adapt block size during runtime.

To address these limitations, we analyze how confidence scores evolve across decoding steps. Our statistical analysis identifies the volatility band (VB), a region in which confidence exhibits high temporal variance over time. We find that VBs often coincide with local semantic units, providing a signal that can be leveraged to adapt block sizes at inference time. Motivated by this observation, we propose *AdaBlock-dLLM*, a training-free, plug-and-play adaptive block-size scheduler for semi-AR dLLM inference that adjusts block boundaries in a semantics-aware manner (Figure 1, right). Specifically, *AdaBlock-dLLM* aligns block size with semantic boundaries, indicated by semantic delimiter tokens (e.g., periods and \n), thereby improving sampling quality. Based on comprehensive experiments on various benchmarks, we demonstrate that *AdaBlock-dLLM* improves accuracy by up to $5.3\%$ while achieving throughput comparable to prior dLLM acceleration methods (Figure 2). Gains are especially pronounced under KV caching, where fixed block sizes further compromise semantic consistency. Our results motivate exploring semantics-aware training objectives for diffusion language models that better preserve contextual coherence.

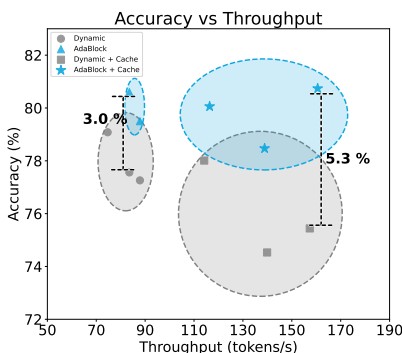

Figure 2: Performance improvement over Fast-dLLM (Wu et al., 2025).

In summary, our contributions are threefold:

- We systematically analyze the semi-autoregressive decoding paradigm, and identify the inaccuracy and inefficiency behind fixed block size settings (Section 4.1 and Section 4.2).
- We propose *AdaBlock-dLLM*, a training-free, plug-and-play technique that enhances the existing semi-autoregressive decoding paradigm, which dynamically adjusts block sizes based on the confidence of semantic delimiter tokens (Section 4.3).
- We conduct extensive experiments demonstrating that *AdaBlock-dLLM* improves accuracy by up to $5.3\%$ over state-of-the-art methods under the same speed budget (Section 5).

## 2 RELATED WORK

### 2.1 DIFFUSION LANGUAGE MODELS

Diffusion models (Sohl-Dickstein et al., 2015; Ho et al., 2020; Song et al., 2021; Karras et al., 2022) have achieved high-fidelity generation across continuous data domains, including images and

video (Peebles & Xie, 2023; Ho et al., 2022). Motivated by this success, a growing line of work adapts diffusion to NLP tasks, giving rise to masked diffusion models (MDMs) that iteratively denoise an initially masked token sequence into coherent output (Austin et al., 2021a; Hoogeboom et al., 2021; Lou et al., 2024). Recent efforts have scaled MDMs up to 8B parameters (Nie et al., 2025; Ye et al., 2025), highlighting their robustness and scalability. Current diffusion language models can be categorized into *i)* models trained from scratch (Nie et al., 2025; Yang et al., 2025), *ii)* models adapted from autoregressive (AR) models (Ye et al., 2025), and *iii)* block-diffusion models (Cheng et al., 2025; Wang et al., 2025d), which combine the training efficiency of AR models with the sampling efficiency of diffusion models.

## 2.2 INFERENCE-TIME OPTIMIZATIONS FOR DIFFUSION LLMS

Recent work investigates inference-time optimization for diffusion LLMs, focusing on inference acceleration and generation quality improvement. For acceleration, existing methods largely fall into two categories: approximate caching and parallel decoding. Caching approaches reuse key–value (KV) states across denoising steps to reduce redundant computation, including delayed caching (Ma et al., 2025), similarity-guided caching (Liu et al., 2025), and block-wise caching (Wu et al., 2025) tailored to semi-AR decoding. Parallel decoding methods improve the speed–quality trade-off via confidence-based dynamic unmasking (Wu et al., 2025; Wang et al., 2025c; Wei et al., 2025), unmasking schedule design (Luxembourg et al., 2025), and guided diffusion (Hu et al., 2025; Israel et al., 2025). Beyond speed, test-time strategies such as voting (Wang et al., 2025b), early commit decoding (Li et al., 2025a), and remasking-based refinement (Hong et al., 2025; Wang et al., 2025a; He et al., 2025) have also been explored to improve generation quality.

## 2.3 BLOCK-WISE SEMI-AUTOREGRESSIVE DECODING

Semi-autoregressive (semi-AR) decoding partitions the sequence into blocks. Decoding is autoregressive over blocks, but non-autoregressive within each block, enabling tokens in the same block to be decoded in arbitrary order. Block Diffusion (Arriola et al., 2025) first introduced this paradigm for dLLMs by interpolating between token-level autoregression and fully non-autoregressive diffusion via a block-causal attention mask. Semi-AR decoding has been commonly adopted by dLLMs, including LLaDA (Nie et al., 2025) and MMaDA (Yang et al., 2025). In prior work, semi-AR decoding typically uses a **fixed** block size. In contrast, this paper takes the first attempt to explore **adaptive** block-size decoding with a **semantic-aware**, **training-free** method.

## 3 PRELIMINARIES

The decoding process of diffusion-based LLMs initializes from a fully masked sequence. Each decoding step alternates between two operations: *denoise*, which predicts token distributions conditioned on the current partially masked sequence, and *sample*, which uses these distributions to unmask masked positions. This denoise–sample cycle is repeated until all positions are unmasked. Using LLaDA (Nie et al., 2025) as an example, we formalize the decoding process below.

**Setup.** Let $\mathcal{V}$ denote the vocabulary, which includes a special mask token $[\texttt{MASK}] \in \mathcal{V}$. Given a prompt $\mathbf{q} = (q_0, \ldots q_{L_p-1}) \in \mathcal{V}^{L_p}$ and a generation budget $L$, define the index set $\mathcal{J} \triangleq \{0, 1, \ldots, L_p + L - 1\}$. Let $T$ be the total number of denoise–sample iterations.

At step $t \in \{T, T-1, \ldots, 0\}$, the sequence state is $\mathbf{y}^t = (y_i^t)_{i \in \mathcal{J}} \in \mathcal{V}^{L_p + L}$. The initial state of the sequence is $\mathbf{y}^T = (q_0, \ldots, q_{L_p-1}, \underbrace{[\texttt{MASK}], \ldots, [\texttt{MASK}]}_{L \text{ times}})$.

**Denoise.** A mask predictor $p_\theta$ predicts a sequence $\hat{\mathbf{y}}^t \in (\mathcal{V} \setminus [\texttt{MASK}])^{L_p + L}$ using greedy decoding:

$$\hat{y}_i^t = \arg\max_{v \in \mathcal{V}} p_\theta(v \mid \mathbf{y}^t, i), \qquad i \in \mathcal{J}. \tag{1}$$

**Sample.** Define the masked-position set:

$$\mathcal{M}_t \triangleq \{ i \in \mathcal{J} : y_i^t = [\texttt{MASK}] \}. \tag{2}$$

A sampler selects $S_t \subseteq \mathcal{M}_t$ to unmask. For all $i \in \mathcal{J}$, update

$$y_i^{t-1} = \begin{cases} \hat{y}_i^t, & i \in S_t \quad \text{(unmask)}, \\ [\texttt{MASK}], & i \in \mathcal{M}_t \setminus S_t \quad \text{(stay masked)}, \\ y_i^t, & i \notin \mathcal{M}_t \quad \text{(already unmasked; keep)}. \end{cases} \tag{3}$$

Then, the sequence state becomes

$$\mathbf{y}^{t-1} = (y_i^{t-1})_{i \in \mathcal{J}} \in \mathcal{V}^{L_p+L}. \tag{4}$$

This process repeats for $t = T, T-1, \ldots, 1$ and terminates when $\mathcal{M}_t = \varnothing$. A

Samplers differ primarily in their unmasking schedules and position-selection rules. LLaDA (Nie et al., 2025) proposes a linear-schedule sampler that unmasks a fixed number of tokens $L/T$ at each step, selecting positions either uniformly at random or by confidence. LaViDa (Li et al., 2025b) applies a shifting schedule that unmasks a variable number of [$\texttt{MASK}$] positions at each step. Fast-dLLM (Wu et al., 2025) and Dimple (Yu et al., 2025) implement dynamic sampling by introducing a confidence threshold $\tau$: at each step $t$, the model unmasks all positions with confidence $c_i^t \triangleq p_\theta(\hat{y}_i^t \mid \mathbf{y}^t, i)$ that $c_i^t \geq \tau$. Compared to the linear-schedule or the shifting schedule, dynamic sampling adapts to token-level uncertainty and improves the accuracy–throughput balance.

## 4 METHODOLOGY

This work focuses on a widely adopted inference paradigm for diffusion-based LLMs: semi-autoregressive (semi-AR) decoding with dynamic sampling (Wu et al., 2025). This paradigm enforces block-causal dependencies, enabling block-wise KV caching and allowing multiple tokens to be decoded within a single denoise–sample iteration. It is governed by two key hyperparameters: the **confidence threshold** $\tau$ and the **block size** $B$. The threshold $\tau$ controls the speed–quality trade-off, whereas the impact of $B$ is yet to be explored. We systematically analyze the drawbacks of fixed block sizes and propose a lightweight sampler that improves sampling quality for dLLMs.

In Section 4.1, we analyze the confidence dynamics that govern dynamic sampling. Using these patterns, we characterize the inaccuracies and inefficiencies caused by the misalignment between a fixed block size and the model's inductive decoding preferences in Section 4.2. This motivates an adaptive block-size scheduler introduced in Section 4.3 that minimizes this mismatch.

### 4.1 ANALYSIS OF CONFIDENCE DYNAMICS

**Confidence Dynamics.** Confidence scores are a key metric in dynamic sampling of dLLM. A high confidence score indicates that the model is more certain about the prediction (Wei et al., 2025). Figure 3 visualizes confidence dynamics at early, middle, and late decoding stages of LLaDA-8B-Base inference on GSM8K Benchmark. Based on these statistical analyses, we summarize the following observations and patterns for different decoding stages:

- For all stages, a high-confidence region emerges near the decoded tokens. This indicates that masked positions adjacent to decoded tokens are more likely to attain high confidence and thus to be decoded. We attribute this to **confidence locality**: dLLMs exhibit higher confidence in regions where local semantic meaning is complete.

- Although the model may generate in arbitrary order, its decoding trace shows a **global autoregressive tendency**. This pattern is consistent with a chain-like progression driven by semantic dependencies, where subsequent predictions rely on prior semantics.

- As the decoding process unfolds, confidence for decoded tokens remains high, and the high-confidence region extends toward adjacent positions. In contrast, positions outside this region maintain consistently low confidence.

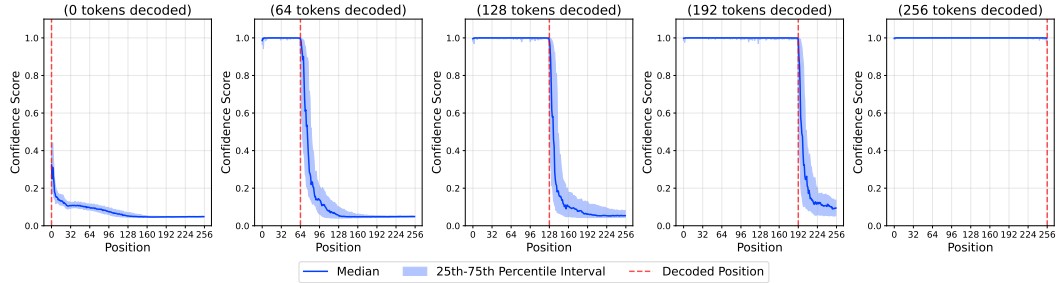

Figure 3: Confidence scores across sequence positions for LLaDA-8B-Base, evaluated on 100 samples from the GSM8K benchmark. The high confidence plateau expands as decoding progresses, while positions adjacent to the decoded prefix exhibit high variance.

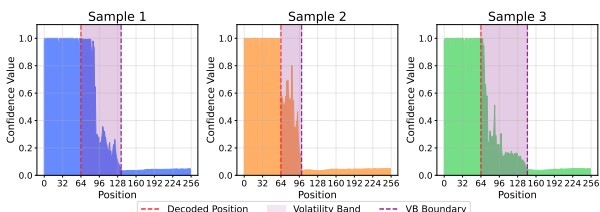

Figure 4: Illustration of the high confidence plateau, the volatility band (VB), and the low confidence floor across three samples. Within VB, the distribution of confidence scores and the width of the band vary across samples.

Figure 5: Proportion of sampling steps affected by late decoding overhead and premature decoding error on GSM8K and HumanEval for fixed block sizes.

**Volatility Band and Local Stochasticity.** Building on these observations, we partition the position-wise confidence landscape into three regions: *i)* a high-confidence plateau with consistently high scores, *ii)* a volatility band (VB) characterized by unstable, variable scores, and *iii)* a low-confidence floor with persistently low scores. Among these regions, the VB is the key region where the current decoding steps take place. As illustrated in Figure 4, scores within the VB fluctuate over decoding steps (time) and across positions (space), yet remain clearly separated from the low-confidence floor. Additionally, the width of the VB varies from case to case. In contrast to the left-to-right expansion of the high-confidence plateau that is driven by global autoregressive causality, the decoding order within the VB is locally stochastic: the positional preference diminishes, and the sampling choice depends more heavily on the immediate semantic context.

## 4.2 MOTIVATION FOR ADAPTIVE BLOCK SIZE.

Despite the local stochasticity exhibited during decoding, prevailing semi-AR decoding strategies use manually set, fixed block sizes that often fail to capture this stochasticity. This misalignment leads to **Late Decoding Overhead** and **Premature Decoding Error**, as illustrated in Figure 1, thereby reducing accuracy and efficiency. This motivates incorporating an adaptive block size.

**Late Decoding Overhead.** A fixed block size poses a hard constraint on the number of tokens that can be sampled in a single step. By fixing the block size, the sampler must exclude nearby higher-confidence positions outside the block, especially when a small block size (Wang et al., 2025d; Cheng et al., 2025) is used. Deferred high-confidence positions undergo additional denoising in later iterations, incurring unnecessary computation overhead and resulting in degraded throughput.

**Premature Decoding Error.** With fixed block sizes, the sampler (Algorithm 3) must decode all masked positions in the current block before proceeding to the next block. Under this constraint, the sampler can be forced to decode low-confidence positions within the block rather than deferring them in favor of higher-confidence positions in the consequent blocks. As a result, the sampler can commit to incorrect tokens early, inducing systematic errors: premature decoding increases token-level error rates, propagates mistakes to subsequent steps by conditioning later blocks on these tokens via block-level autoregressive dependencies, and shifts generation toward poorly calibrated regions of the confidence landscape. This usually contributes to degraded accuracy.

---

**Algorithm 1** Semantic-Aware Block Size Determination

---

**Inputs:** predicted sequence $\hat{y}$; confidences $\mathbf{c}$; generation budget $L$; default block size $B_0$; delimiter set $\mathcal{D}$; delimiter threshold $\tau_{\mathcal{D}}$; current position $g$.

**Output:** block size $B$

1: **function** COMPUTEBLOCKLENGTH($\hat{y}$, $\mathbf{c}$, $L$, $B_0$, $\mathcal{D}$, $\tau_{\mathcal{D}}$, $g$)
2:     ▷ *Sampling window boundary*
3:     $start, remaining \leftarrow g, L - g$
4:     $w \leftarrow \min\big(\max(1, \lfloor 0.25 \cdot g \rfloor), remaining\big)$
5:     $W \leftarrow \{\, start, \dots, start + w - 1 \,\}$                 ▷ window token indices
6:     ▷ *Find highest-confidence delimiter*
7:     $\mathcal{I} \leftarrow \{\, i \in W \mid \hat{y}_i \in \mathcal{D} \,\}$
8:     **if** $\mathcal{I} \neq \emptyset$ **then**
9:         $pos \leftarrow \arg\max_{i \in \mathcal{I}} c_i$          ▷ Select position with max delimiter token confidence
10:        $c_{\max} \leftarrow c_{pos}$
11:     **else**
12:        $c_{\max} \leftarrow -\infty$
13:     **end if**
14:     ▷ *Determine block size*
15:     **if** $c_{\max} \geq \tau_{\mathcal{D}}$ **then**
16:        $B \leftarrow (pos - start + 1)$          ▷ inclusive length up to the delimiter token
17:     **else**
18:        $B \leftarrow \min(B_0, remaining)$
19:     **end if**
20:     **return** $B$
21: **end function**

---

## 4.3   SEMANTIC-AWARE ADAPTIVE BLOCK-SIZE SCHEDULER

**Challenge for predicting block size.** As explained in Section 4.1, the VB delineates the span of positions undergoing active decoding: positions preceding the VB are decoded and exhibit consistently high, stable confidence, whereas positions within the VB show pronounced step-to-step fluctuations. Additionally, Figure 4 shows that the denoised predictions in VB typically relate to the semantic context. In contrast, positions in the low-confidence floor are repeatedly predicted as non-content tokens, such as placeholders or formatting symbols.

Although the VB region inspires scheduling block-sizes adaptively, it often spans too many positions, failing to provide actionable guidance for scheduling. As shown in Figure 7, the token "GB" repeatedly appears with confidence scores mostly between $0.1$ and $0.3$, placing it inside the VB. While such tokens may be contextually related, their significance to the current context is weak, thus offering limited guidance for the current sampling step. Consequently, VB regions alone are often insufficient for local scheduling decisions, motivating a finer-grained segmentation to identify tokens that are most tightly coupled to the current decoding step.

**Aligning Block Size With Semantic Steps.** To obtain a fine-grained segmentation that reflects the context of the current decoding step, we partition the sequence into **semantic steps**, which are contiguous spans whose provisional tokens exhibit local semantic coherence. We then couple the scheduler to the semantic step, setting the block size guided by the length of the current semantic step. This allows for finalizing higher-confidence positions within the step while deferring lower-confidence positions until the semantic step is ready to close. Across semantic steps, dependencies are enforced by the semi-AR paradigm, since each downstream step conditions on completed predecessors. This alignment curbs premature commitments outside the active step and prevents splitting a step across iterations, thereby reducing both error propagation and computational overhead.

**Semantic-Aware Block Size Scheduling.** To facilitate the dynamic scheduling of block size $B$, we insert an additional block-size determination procedure (Algorithm 1) before sampling the first token of each block. This procedure predicts the block size that aligns with the length of semantic steps. Given the current predicted sequence $\hat{y}$ and confidence scores $\mathbf{c}$, we collect indices whose predicted tokens fall in the delimiter set $\mathcal{D}$ (line 7). Tokens in the delimiter set indicate the end of

a semantic unit and demonstrate sharp confidence drops (Appendix A.4). We choose the delimiter token $\hat{y}_{\max}$ with the highest confidence $c_{\max}$. If $c_{\max} \geq \tau_D$, we set $B$ to the position of $\hat{y}_{\max}$ (lines 15–16), indicating that the model has a reliable preference for the end of the current semantic step. If no delimiters appear in $\hat{\mathbf{y}}$ within $W$ (lines 11–12), or if all delimiter predictions are low-confidence ($c_{\max} < \tau_D$, lines 17–18), we fall back to the default block size. Additionally, we apply an index window $W$ that masks distant positions to avoid decoding the `<EOS>` token in the early stage, a cause for severe performance drop (Nie et al., 2025). This procedure yields step-aware blocks when evidence is strong, while remaining conservative in ambiguous regions.

## 5 EXPERIMENT

### 5.1 EXPERIMENTAL SETUP

**Implementation Details.** We evaluate *AdaBlock-dLLM* on representative diffusion LLMs: LLaDA-8B-Instruct (Nie et al., 2025), LLaDA-1.5 (Zhu et al., 2025), and Dream-v0-Base-7B (Ye et al., 2025). Unless otherwise noted, all experiments run on NVIDIA H100 GPUs.

**Hyperparameter Settings.** We use the generation budget $L = 512$ for all benchmarks. For dynamic sampling, we use a confidence threshold $\tau = 0.9$. We sweep default block sizes $B \in \{16, 32, 64\}$. *AdaBlock-dLLM* introduces two hyperparameters: the delimiter set $\mathcal{D}$ and the delimiter confidence threshold $\tau_D$. We set $\mathcal{D} = \{\backslash n\}$, which commonly marks the end of reasoning steps in test-time search (Snell et al., 2024), and frequently causes a significant confidence drop (Appendix A.4). We use $\tau_D = 0.3$ for LLaDA models and $\tau_D = 0.5$ in Dream models. This is tuned on a small subset of the GSM8K benchmark, and the selection is discussed in Section 5.3.

**Benchmarks and Metrics.** We evaluate *AdaBlock-dLLM* on standard LLM benchmarks. For math reasoning, we use GSM8K (5-shot) (Cobbe et al., 2021) and MATH (4-shot) (Hendrycks et al., 2021). For code generation, we use HumanEval (0-shot) (Chen et al., 2021) and MBPP (3-shot) (Austin et al., 2021b). Generation quality is measured by accuracy: pass@1 for code generation and answer accuracy for math reasoning. We compare five sampling methods: **Vanilla** (top-1 confidence sampling); **Dynamic** (threshold-based dynamic sampling) and **+Cache** (**Dynamic** method with DualCache), both following Fast-dLLM (Wu et al., 2025); and two variants enhanced with *AdaBlock-dLLM*, **+Ada** (**Dynamic** method with adaptive block sizing) and **+Ada+Cache** (**+Cache** method with adaptive block sizing).

### 5.2 MAIN RESULTS

**Generation Quality Across Models.** Table 1 reports accuracy to quantify generation quality. *AdaBlock-dLLM* achieves accuracy gains across most model and dataset pairs. Notably, on GSM8K with LLaDA-Instruct, accuracy improves by $3.0\%$ without caching and by $5.3\%$ with caching.

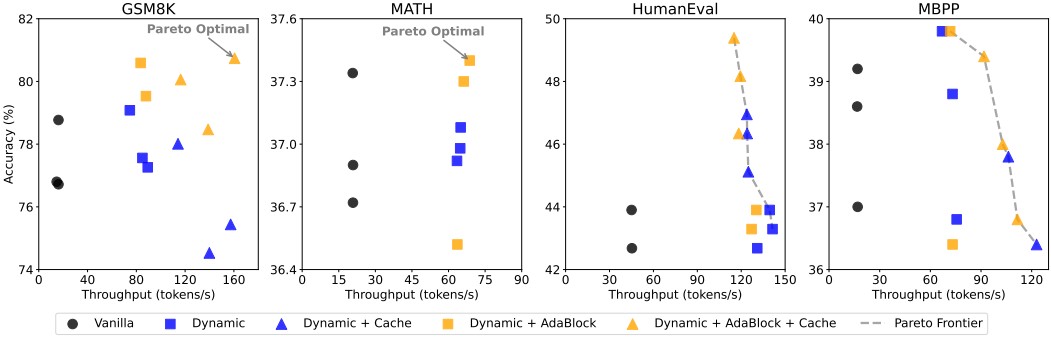

Figure 6: Accuracy and throughput of different sampling methods evaluated on LLaDA-Instruct. Integrating *AdaBlock-dLLM* into Fast-dLLM (Wu et al., 2025) yields accuracy gains across all benchmarks while maintaining little throughput overhead. Notably, Fast-dLLM with *AdaBlock-dLLM* is Pareto-optimal on the GSM8K and MATH datasets.

Table 1: Accuracy (%) across sampling methods, evaluated on LLaDA-1.5, LLaDA-Instruct, and Dream-Base under default block sizes $B_0 \in \{16, 32, 64\}$. Differences are shown in gray. Comparisons are reported relative to **Dynamic** and **+Cache** (Wu et al., 2025).

| Method | LLaDA-Instruct | | | LLaDA-1.5 | | | Dream-Base | | |
|---|---|---|---|---|---|---|---|---|---|
| | $B_0 = 16$ | $B_0 = 32$ | $B_0 = 64$ | $B_0 = 16$ | $B_0 = 32$ | $B_0 = 64$ | $B_0 = 16$ | $B_0 = 32$ | $B_0 = 64$ |
| **GSM8K** | | | | | | | | | |
| Vanilla | 78.8 | 76.7 | 76.8 | 82.3 | 82.3 | 80.4 | 76.3 | 76.4 | 75.1 |
| Dynamic | 79.1 | 77.6 | 77.3 | 82.6 | 82.2 | 80.7 | 75.5 | 75.5 | 75.6 |
| +Ada | 80.6 +1.5 | 80.6 +3.0 | 79.5 +2.2 | 83.0 +0.4 | 82.4 +0.2 | 80.3 -0.4 | 75.7 +0.2 | 75.7 +0.2 | 75.9 +0.3 |
| +Cache | 78.0 | 74.5 | 75.4 | 80.7 | 80.2 | 80.0 | 75.6 | 74.5 | 74.6 |
| +Ada+Cache | 80.0 +2.0 | 78.5 +4.0 | 80.7 +5.3 | 81.3 +0.6 | 81.7 +1.5 | 79.7 -0.3 | 76.5 +0.9 | 75.1 +0.6 | 74.6 +0.0 |
| **HumanEval** | | | | | | | | | |
| Vanilla | 43.9 | 43.9 | 42.7 | 39.0 | 36.6 | 38.4 | 53.7 | 52.4 | 54.3 |
| Dynamic | 42.7 | 43.9 | 43.3 | 36.6 | 37.8 | 36.6 | 53.0 | 51.2 | 52.4 |
| +Ada | 43.3 +0.6 | 43.3 -0.6 | 43.9 +0.6 | 37.8 +1.2 | 38.4 +0.6 | 38.4 +1.8 | 53.7 +0.7 | 51.2 +0.0 | 53.7 +1.3 |
| +Cache | 45.1 | 46.3 | 47.0 | 33.5 | 36.0 | 34.1 | 50.0 | 53.0 | 56.1 |
| +Ada+Cache | 49.4 +4.3 | 46.3 +0.0 | 48.2 +1.2 | 36.0 +2.5 | 39.0 +3.0 | 36.0 +1.9 | 52.4 +2.4 | 53.0 +0.0 | 57.3 +1.2 |
| **MATH** | | | | | | | | | |
| Vanilla | 36.7 | 36.9 | 37.3 | 36.3 | 37.0 | 34.4 | 39.8 | 40.2 | 40.1 |
| Dynamic | 37.0 | 36.9 | 37.1 | 36.3 | 36.7 | 34.4 | 39.7 | 39.9 | 39.9 |
| +Ada | 36.5 -0.5 | 37.3 +0.4 | 37.4 +0.3 | 36.8 +0.5 | 36.7 +0.0 | 34.1 -0.3 | 39.6 -0.1 | 39.9 +0.0 | 39.9 +0.0 |
| +Cache | 35.4 | 35.8 | 36.0 | 34.9 | 33.2 | 32.1 | 38.0 | 38.5 | 38.8 |
| +Ada+Cache | 35.8 +0.4 | 35.3 -0.5 | 35.6 -0.4 | 35.2 +0.3 | 33.9 +0.7 | 32.4 +0.3 | 37.8 -0.2 | 38.4 -0.1 | 38.4 -0.4 |
| **MBPP** | | | | | | | | | |
| Vanilla | 39.2 | 38.6 | 37.0 | 38.2 | 37.0 | 23.2 | 12.4 | 12.4 | 12.8 |
| Dynamic | 39.8 | 38.8 | 36.8 | 38.2 | 37.0 | 24.6 | 12.6 | 12.4 | 12.2 |
| +Ada | 40.2 +0.4 | 39.8 +1.0 | 36.4 -0.4 | 39.4 +1.2 | 37.6 +0.6 | 29.8 +5.2 | 12.8 +0.2 | 14.2 +1.8 | 12.4 +0.2 |
| +Cache | 35.6 | 37.8 | 36.4 | 38.0 | 34.8 | 19.8 | 12.8 | 11.6 | 9.6 |
| +Ada+Cache | 39.4 +3.8 | 38.0 +0.2 | 36.8 +0.4 | 36.6 -1.4 | 36.4 +1.6 | 36.6 +6.8 | 12.8 +0.0 | 11.6 +0.0 | 12.4 +2.8 |

**Pronounced Accuracy Improvement With Cache.** We observe that accuracy gains are particularly pronounced when KV caching is applied. Unlike autoregressive decoding, where caching is effectively lossless, block-wise KV caching in dLLMs is an approximation because key and value tensors vary across time steps, and the decoding order within each block is non-sequential. This approximation results in notable accuracy degradation at large block sizes.

Table 3 reports accuracy gains with both PrefixCache and DualCache (Wu et al., 2025). Improvements are twofold: *i)* For large default block sizes $B_0$, the resulting average block size $\bar{B}$ is smaller, reducing the approximation error of KV Caching. *ii)* By enhancing semantic locality within each block, inter-block dependencies are reduced, making decoding less sensitive to stale cached tensors. The second effect is more impactful: on GSM8K, $\bar{B}$ for $B_0 = 64$ is 33.98, yet accuracy still exceeds that of $B_0 = 32$ without *AdaBlock-dLLM* by $1.90\%$ (no cache) and $6.20\%$ (DualCache). Given that block-wise KV caching is a core advantage of semi-AR decoding, these results indicate that the method integrates seamlessly with existing techniques that improve inference efficiency.

**Discussion on Difference Between Models.** The gains from *AdaBlock-dLLM* vary across models. In particular, *AdaBlock-dLLM* yields larger improvements on the LLaDA family. By predicting block size informed by local semantics, *AdaBlock-dLLM* groups tokens that constitute a semantic step into the same block and focuses on refining the local context. This effect is strongest for dLLMs trained from scratch, which exhibit greater local stochasticity. In contrast, dLLMs adapted from AR models display global autoregressive order and a high degree of local autoregressiveness (Gong et al., 2025). In such a scenario, the improvements from *AdaBlock-dLLM* are correspondingly smaller. Additionally, the generation quality is also limited by the model's denoising quality, while our work focuses on mitigating the performance loss of sampling within fixed-sized blocks.

**Discussion on Throughput Overhead.** Table 2 reports the accuracy, throughput (measured in tokens per second, TPS), and the average number of function evaluations (NFE). The product of throughput and NFE remains stable across methods and block sizes, indicating an approximately

Table 2: Performance comparison across default block sizes $B_0$ under a generation budget of $L = 512$. The product of throughput and NFE is nearly identical across methods and block sizes, indicating an approximately inverse relationship between these quantities when no block-wise KV caching is applied. *AdaBlock-dLLM* yields throughput gains for $B_0 \in \{4, 8\}$. Boldface indicates superior performance; this convention applies to all tables unless noted otherwise.

| | Acc. (%) | TPS | Avg. NFE | TPS×NFE ($10^3$) | Acc. (%) | TPS | Avg. NFE | TPS×NFE ($10^3$) | Acc. (%) | TPS | Avg. NFE | TPS×NFE ($10^3$) |
|---|---|---|---|---|---|---|---|---|---|---|---|---|
| **Method** | | $B_0 = 4$ | | | | $B_0 = 8$ | | | | $B_0 = 16$ | | |
| Vanilla | 80.9 | 16.1 | 512.0 | 8.2 | 80.5 | 16.1 | 512.0 | 8.2 | 78.8 | 16.1 | 512.0 | 8.2 |
| Dynamic | 81.2 | 43.0 | 189.4 | 8.2 | 80.6 | 60.0 | 135.5 | 8.1 | 79.1 | **74.7** | **109.2** | 8.2 |
| +Ada | **81.6** | **51.3** | 159.8 | 8.2 | **81.8** | **63.9** | 128.3 | 8.2 | **80.6** | 73.9 | 102.4 | 8.2 |
| **Method** | | $B_0 = 32$ | | | | $B_0 = 64$ | | | | $B_0 = 128$ | | |
| Vanilla | 76.8 | 16.1 | 512.0 | 8.2 | 76.8 | 16.1 | 512.0 | 8.2 | 71.0 | 16.0 | 512.0 | 8.2 |
| Dynamic | 77.6 | **85.0** | 94.9 | 8.1 | 77.3 | **89.4** | 91.6 | 8.2 | 70.7 | **81.2** | 101.2 | 8.2 |
| +Ada | **80.6** | 83.5 | 98.5 | 8.2 | **79.5** | 87.9 | 93.4 | 8.2 | **72.8** | 80.6 | 101.6 | 8.2 |

Table 3: Accuracy (%) on GSM8K for LLaDA-Instruct across caching methods with $L = 512$.

| Method | $B_0 = 16$ | $B_0 = 32$ | $B_0 = 64$ |
|---|---|---|---|
| +PrefixCache | 78.2 | 76.9 | 75.0 |
| **+Ada+PrefixCache** | **81.4** | **79.8** | **77.6** |
| +DualCache | 78.0 | 74.5 | 75.4 |
| **+Ada+DualCache** | **80.0** | **78.5** | **80.7** |

Table 4: Accuracy (%) on GSM8K for LLaDA-Instruct under different generation budgets.

| Method | $L = 256$ | $L = 512$ | $L = 1024$ |
|---|---|---|---|
| Dynamic | 78.1 | 77.6 | 77.4 |
| **+Ada** | **78.5** | **80.6** | **79.3** |
| +Cache | 77.4 | 74.5 | 75.8 |
| **+Ada+Cache** | **79.2** | **78.5** | **78.1** |

inverse relationship between these metrics. This observation suggests that throughput can be improved primarily by reducing NFE, for example, by increasing the parallelism capacity of the denoiser (Wang et al., 2025c) or by improving the sampling efficiency of the sampler.

We observe that vanilla decoding, which fixes NFE to the generation budget, maintains almost identical throughput across block sizes. In contrast, dynamic sampling with either fixed or adaptive block sizes benefits from increasing $B_0$ from 4 to 64, but throughput drops when $B_0$ is increased further. We attribute this initial positive correlation between block size and throughput to the mitigation of late decoding overhead. The subsequent degradation in throughput arises from less reliable denoiser predictions under large blocks: as $B_0$ grows, within-block semantic coherence weakens, fewer tokens meet the sampling criterion per step, and dynamic sampling therefore decodes fewer tokens per iteration, increasing NFE. Since throughput is approximately inversely proportional to NFE when per-iteration cost is nearly constant, the higher NFE reduces throughput. This effect is consistent with the accuracy trend, where a large block size ($B_0 = 128$) exhibits a marked performance drop.

With *AdaBlock-dLLM*, throughput increases for small default block sizes ($B_0 \in \{4, 8\}$) and decreases slightly for larger defaults ($B_0 \in \{16, 32, 64, 128\}$). *AdaBlock-dLLM* attempts to align the block size with the length of semantic steps, which yields $B > B_0$ on average when the default is small and $B < B_0$ on average when the default is large. For small $B_0$, *AdaBlock-dLLM* effectively reduces late decoding overhead. As $B_0$ increases, selecting $B < B_0$ strengthens within-block coherence and typically improves sampling quality, at the cost of a modest throughput reduction. Nevertheless, **Dynamic+Ada** consistently achieves higher accuracy than **Dynamic** while maintaining comparable throughput. Figure 6 presents the trade-off between accuracy and throughput, demonstrating the improvements induced by our method.

## 5.3 Ablation Studies

**Performance Across Different Generation Budgets.** We evaluate *AdaBlock-dLLM* under three generation budgets: $L \in \{256, 512, 1024\}$, as shown in Table 4. Across all generation budgets $L$ and decoding settings, *AdaBlock-dLLM* improves generation quality. These results motivate a semantics-aware block-size scheduling design for dLLMs.

**Effect on Delimiter Threshold $\tau_D$.** We evaluate three delimiter thresholds for each model family, as shown in Table 5. We observe that $\tau_D = 0.3$ yields the best performance in most cases for

Table 5: Accuracy (%) on GSM8K for LLaDA and Dream across delimiter thresholds $\tau_D \in \{0.3, 0.5, 0.7\}$ with $B_0 = 32$. A smaller $\tau_D$ provides sufficient semantic guidance for dLLMs trained from scratch (e.g., LLaDA).

| Model | $\tau_D = 0.3$ | $\tau_D = 0.5$ | $\tau_D = 0.7$ |
|---|---|---|---|
| LLaDA-Instruct | **80.59** | 79.08 | 77.94 |
| Dream-Base | 75.66 | **75.74** | **75.74** |

Table 6: Accuracy (%) on IFEval for LLaDA-1.5 across sampling methods. *AdaBlock-dLLM* also improves performance.

| Method | $B_0 = 16$ | $B_0 = 32$ | $B_0 = 64$ |
|---|---|---|---|
| Vanilla | 69.1 | 66.7 | 61.3 |
| Dynamic | 69.0 | 66.7 | 61.2 |
| +Ada | 68.4 -0.6 | 67.5 +0.8 | 64.4 +3.2 |
| +Cache | 67.5 | 64.6 | 59.4 |
| +Ada+Cache | 68.9 +1.4 | 66.4 +1.8 | 62.7 +3.3 |

Table 7: Accuracy (%) on GSM8K across eight different delimiter sets with $B_0 = 32$. Results show that using the newline token (\n) as the delimiter accounts for most of the accuracy gains, while additionally including the comma and period further improves performance.

| Delimiter Set | Acc. (%) |
|---|---|
| None (+Cache) | 74.5 |
| {[\n]} | **78.5** |
| {[,]} | 75.1 |
| {[.]} | 74.5 |
| {[,], [.]} | 75.1 |
| {[\n], [,]} | **78.5** |
| {[\n], [.]} | **78.3** |
| {[\n], [,], [.]} | **78.7** |

LLaDA, whereas a higher $\tau_D = 0.5$ is optimal for Dream. We attribute this difference to the distinct confidence distributions of the two models. LLaDA is trained purely from scratch and exhibits lower variance within the volatility band, whereas Dream is adapted from autoregressive models and shows substantially higher variance (Figure 8). Consequently, different thresholds are required to track the boundary of a semantic step. Additionally, overly high thresholds (e.g., $\tau_D = 0.9$) often cause the scheduler to revert to its default behavior, reducing its effectiveness.

**Selection of Delimiter Set $\mathcal{D}$.** We apply more delimiter sets $\mathcal{D}$ to include additional tokens (comma and period), which often mark the termination of local semantic context. Table 7 shows that although accuracy improvements vary, the inclusion of the comma and period achieves higher accuracy than the dynamic sampling baseline. These results highlight the importance of aligning block size with semantic steps in semi-AR decoding. We provide further analysis in A.4.

**Performance on Non-Reasoning Benchmarks.** We further evaluate the performance of *AdaBlock-dLLM* on IFEval (Zhou et al., 2023), a benchmark that examines the instruction-following capability of LLMs. As shown in Table 6, with the delimiter set $\mathcal{D} = \{[\n], [,], [.]\}$, *AdaBlock-dLLM* yields accuracy improvements, especially when integrated with block-wise KV caching. These results suggest that aligning block sizes with semantic steps effectively enhances sampling quality, leading to improved performance on tasks other than math reasoning or coding.

**Limitations.** *AdaBlock-dLLM* effectively augments the existing semi-AR decoding paradigm by aligning block sizes with semantic steps. However, the proposed block scheduler may be less effective when the generation budget is small (e.g., multiple-choice questions), where semi-AR decoding is not particularly beneficial. Moreover, dLLM decoding alternates denoising and sampling, and *AdaBlock-dLLM* primarily improves sampling quality. When the denoiser's predicted token distributions are unreliable, adaptive block sizing cannot recover quality, and the advantage of *AdaBlock-dLLM* shrinks. Future work also includes automating the choice of delimiter tokens and their associated delimiter threshold, rather than relying on fixed heuristics.

## 6 CONCLUSION

This work proposes *AdaBlock-dLLM*, a training-free, plug-and-play scheduler that enhances the existing semi-autoregressive decoding paradigm. We identify two fundamental limitations of conventional semi-AR decoding (late decoding overhead and premature decoding error) that motivate adaptive block-size scheduling. Building on an analysis of confidence dynamics, *AdaBlock-dLLM* adaptively adjusts the block size at runtime, aligning block sizes with semantic steps. Extensive experiments across benchmarks demonstrate improvements in generation quality of up to $5.3\%$ under a comparable throughput budget. We hope that our semantic-aware adaptive approach and statistical analysis will inspire future training and inference strategies for dLLMs.

## ETHICS STATEMENT

This work adheres to the ICLR Code of Ethics. In this study, no research involving human subjects or animals was conducted. All datasets used, including GSM8K, MATH, HumanEval, and MBPP, were obtained in compliance with relevant usage guidelines, ensuring no violations of privacy. We took care to avoid the bias and discriminatory outcomes throughout our research process. No personally identifiable information was used, and no experiments were conducted that could raise privacy or security concerns. We are committed to maintaining transparency and integrity throughout the research process.

## REPRODUCIBILITY STATEMENT

We ensure that all reported results are reproducible. The code repository is open-sourced to facilitate replication and verification. The experimental setup, including model configurations and hardware details, is described in Section 5.1. We also provide a full description of our methodology, including detailed pseudocode in Algorithm 1, to support understanding and reproduction. Additionally, the open-source models used in this work (e.g., LLaDA and Dream) are publicly available, enabling consistent and reproducible evaluation.

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

# A APPENDIX

## A.1 CASE STUDY FOR THE TWO FUNDAMENTAL ISSUES IN SEMI-AUTOREGRESSIVE DECODING

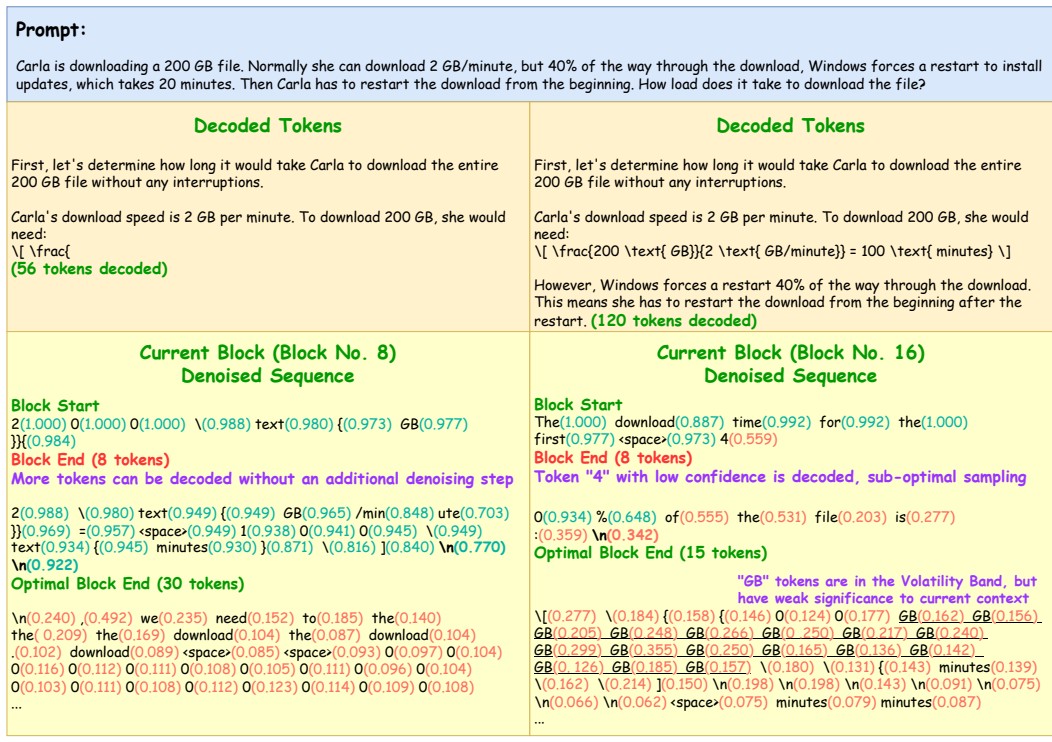

Figure 7: A case study of the two fundamental issues (Late Decoding Overhead and Premature Decoding Error). The configuration uses dynamic sampling with a generation budget of $L = 512$ and a block size of $B = 8$.

## A.2 SEMI-AR DECODING WITH ADAPTIVE BLOCK-SIZE SCHEDULING ALGORITHM

---

**Algorithm 2** Auxiliary: Denoiser

**Inputs:** mask predictor $p_\theta$; vocabulary $\mathcal{V}$; index set $\mathcal{J}$; current sequence $\mathbf{y}$.
**Outputs:** predicted sequence $\hat{\mathbf{y}}$; confidences $\mathbf{c}$
1: **function** DENOISER($p_\theta$, $\mathcal{V}$, $\mathcal{J}$, $\mathbf{y}$)
2:     **for** $i \in \mathcal{J}$ **do**
3:         $\hat{y}_i \leftarrow \arg\max_{v \in \mathcal{V}} \ p_\theta(v \mid \mathbf{y}, i)$
4:         $c_i \leftarrow \max_{v \in \mathcal{V}} \ p_\theta(v \mid \mathbf{y}, i)$
5:     **end for**
6:     **return** $\hat{\mathbf{y}}$, $\mathbf{c}$
7: **end function**

---

---

**Algorithm 3** Auxiliary: Threshold-Based Dynamic Sampling

---

**Inputs:** current sequence $\mathbf{y}$; predicted sequence $\hat{\mathbf{y}}$; confidences $\mathbf{c}$; unmasking threshold $\tau$; index set $\mathcal{J}$.

**Output:** sampled index set $\mathcal{S}$

1: **function** THRESHOLD-SAMPLE($\mathbf{y}$, $\hat{\mathbf{y}}$, $\mathbf{c}$, $\tau$, $\mathcal{J}$)
2:     $\mathcal{S} \leftarrow \emptyset$
3:     $\mathcal{J}_{\text{mask}} \leftarrow \{\, i \in \mathcal{J} \mid y_i = \texttt{[MASK]} \,\}$         ▷ Select all masked positions in the block
4:     **Select** index $i_{\text{top}}$ with highest confidence $c_i$
5:     $\mathcal{S} \leftarrow S \cup i_{\text{top}}$
6:     **for** $i \in \mathcal{J}$ **do**
7:         **if** $i \in \mathcal{J}_{\text{mask}} \,\wedge\, c_i \geq \tau$ **then**
8:             $\mathcal{S} \leftarrow \mathcal{S} \cup \{i\}$
9:         **end if**
10:     **end for**
11:     **return** $\mathcal{S}$
12: **end function**

---

**Algorithm 4** Semi-AR Decoding with Adaptive Block-Size Scheduling

---

**Inputs:** mask predictor $p_\theta$; vocabulary $\mathcal{V}$; initial sequence $\mathbf{y}^{(T)}$ with index set $\mathcal{J}$; generation budget $L$; default block size $B_0$; delimiter set $\mathcal{D}$; delimiter threshold $\tau_{\mathcal{D}}$; unmasking threshold $\tau$.

**Output:** decoded sequence $\mathbf{y}$.

1: generated length $g \leftarrow 0$,   timestep $t \leftarrow T$
2: **while** $g < L \,\wedge\, t \geq 1$ **do**
3:
4:     ▷ *First denoising to obtain predicted sequence and confidence at step $t$*
5:     $(\hat{\mathbf{y}}^{(t)}, \mathbf{c}^{(t)}) \leftarrow$ DENOISER$(p_\theta, \mathcal{V}, \mathcal{J}, \mathbf{y}^{(t)})$
6:
7:     ▷ *Compute block size*
8:     $B \leftarrow$ COMPUTEBLOCKLENGTH$(\hat{\mathbf{y}}^{(t)}, \mathbf{c}^{(t)}, L, B_0, \mathcal{D}, \tau_{\mathcal{D}}, g)$
9:     $\mathcal{J}_{\text{blk}} \leftarrow \{\, g, g+1, \ldots, g+B-1 \,\}$
10:
11:     ▷ *First sample*
12:     $\mathcal{S} \leftarrow$ THRESHOLD-SAMPLE$(\mathbf{y}^{(t)}, \hat{\mathbf{y}}^{(t)}, \mathbf{c}^{(t)}, \tau, \mathcal{J}_{\text{blk}})$
13:     **for** $i \in \mathcal{S}$ **do**
14:         $y_i^{t-1} \leftarrow \hat{y}_i^t$         ▷ sample tokens with high confidence
15:     **end for**
16:
17:     $y_j^{(t-1)} \leftarrow y_j^{(t)} \;\; \forall j \notin \mathcal{S}, \quad t \leftarrow t - 1$         ▷ Copy other tokens
18:     $\mathcal{J}_{\text{mask}} \leftarrow \{\, i \in \mathcal{J}_{\text{blk}} \mid y_i^{(t)} = \texttt{[MASK]} \,\}$
19:
20:     ▷ *In-block denoise–sample cycles*
21:     **while** $\mathcal{J}_{\text{mask}} \neq \emptyset \,\wedge\, t \geq 1$ **do**
22:         $(\hat{\mathbf{y}}^{(t)}, \mathbf{c}^{(t)}) \leftarrow$ DENOISER$(p_\theta, \mathcal{V}, \mathcal{J}_{\text{mask}}, \mathbf{y}^t)$
23:         $\mathcal{S} \leftarrow$ THRESHOLD-SAMPLE$(\mathbf{y}^t, \hat{\mathbf{y}}^{(t)}, \mathbf{c}^{(t)}, \tau, \mathcal{J}_{\text{mask}})$
24:         **for** $i \in \mathcal{S}$ **do**
25:             $y_i^{(t-1)} \leftarrow \hat{y}_i^{(t)}$
26:         **end for**
27:         $y_j^{(t-1)} \leftarrow y_j^{(t)} \;\; \forall j \notin \mathcal{S}, \quad t \leftarrow t - 1$
28:         $\mathcal{J}_{\text{mask}} \leftarrow \{\, i \in \mathcal{J}_{\text{blk}} \mid y_i^{(t)} = \texttt{[MASK]} \,\}$
29:     **end while**
30:
31:     $g \leftarrow g + B$
32: **end while**
33:
34: **return** $\mathbf{y}^t$

---

## A.3 Confidence Dynamics for Dream-Base

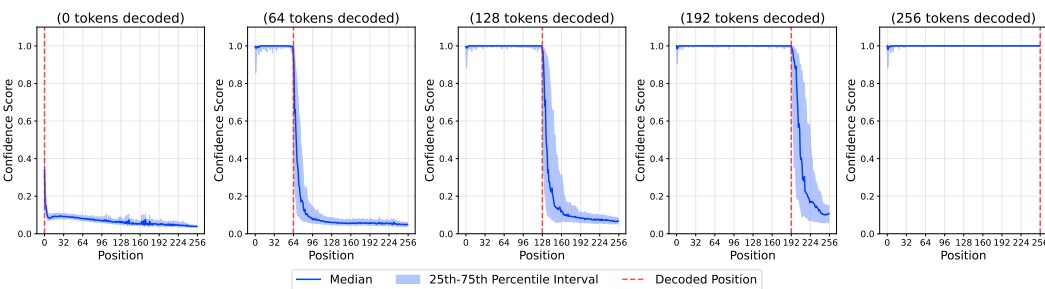

Figure 8: Confidence scores across sequence positions for Dream-v0-Base-7B, evaluated on 100 samples from the GSM8K benchmark. Adapted models such as Dream exhibit a similar degree of global autoregressiveness to LLaDA, but with higher variance in the volatility band. This increased variance motivates the use of a higher delimiter threshold $\tau_D$ to provide stronger semantic guidance.

## A.4 Further Analysis of the Delimiter Set

The choice of \n as a delimiter token is supported by statistics of confidence drops between consecutive positions. We measure a confidence drop as the decrease in token-level confidence from one position to the next. Large drops within the volatility band are correlated with semantic transitions and thus provide a heuristic for segmenting semantic steps. In a sample of 100 GSM8K examples, \n is the token that most frequently leads to large confidence drops, consistent with its role as a boundary marker. Further, Table 7 lists additional high-frequency tokens associated with large confidence drops (e.g., commas and periods); incorporating these tokens into the delimiter set improves sampling accuracy in our ablations.

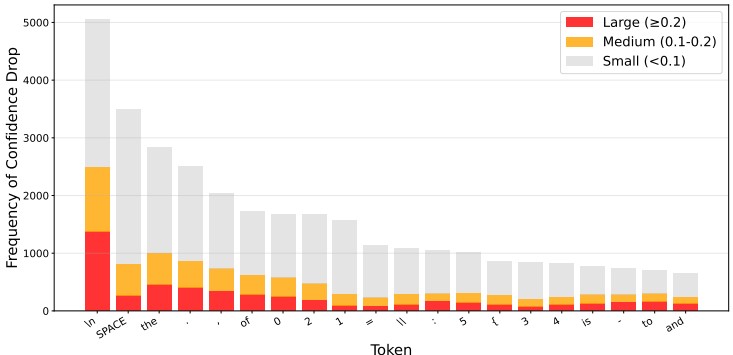

Figure 9: The frequency of confidence drops between consecutive tokens.

## A.5 The Use of Large Language Models

LLMs were used for polishing the manuscript. Specifically, we used an LLM to assist in correcting grammar errors to improve readability. It is important to note that the LLM was not involved in the ideation, research methodology, or experimental design. All intellectual contributions and scientific ideas developed in this work originated from the authors. The contributions of the LLM were solely focused on improving the linguistic quality of the paper, with no involvement in the scientific content or data analysis. The authors take full responsibility for the content of the manuscript, including any text polished by the LLM. We have ensured that the LLM-polished text adheres to ethical guidelines and does not contribute to plagiarism or scientific misconduct.

