# OpenReview forum: "AdaBlock-dLLM: Semantic-Aware Diffusion LLM Inference via Adaptive Block Size"
_ICLR.cc/2026/Conference — ICLR 2026 Poster_

### Official Review · Reviewer_caBK · 2025-10-29

**Soundness:** 3
**Presentation:** 4
**Contribution:** 3
**Rating:** 8
**Confidence:** 4

**Summary:**

This paper investigates the limitations of fixed block-size semi-autoregressive decoding in diffusion-based large language models (dLLMs). The authors identify two inefficiencies — Late Decoding Overhead and Premature Decoding Error — that arise when fixed-size blocks fail to align with semantic structure during decoding. To address this, they propose AdaBlock-dLLM, a training-free, plug-and-play adaptive scheduler that dynamically adjusts block size according to semantic cues and confidence scores during inference. The method leverages a novel concept called the Volatility Band (VB) — regions of fluctuating confidence that correspond to evolving semantic steps — to determine when to expand or contract blocks.

**Strengths:**

- Well motivated. The paper convincingly articulates the inefficiencies of fixed block-size decoding in diffusion LLMs.

- The adaptive block-size scheduler is conceptually elegant, lightweight, and compatible with existing architectures.

- The authors provide extensive results across multiple models and datasets. The method yields consistent accuracy improvements, particularly when combined with caching.

**Weaknesses:**

- The experiments focus mainly on math  and code generation. Broader text generation tasks (e.g., summarization or translation) and harder math reasoning tasks such as AIME would strengthen claims of generality.

- At larger block sizes, AdaBlock-dLLM shows reduced throughput compared to the baseline, raising concerns about its scalability for high-speed inference.

- The delimiter threshold (τ_D) and delimiter set (D) are manually tuned per model family. A more principled or automated way to select these would increase robustness.

- The paper lacks a discussion of situations when adaptive block size might fail.

**Questions:**

Technical Concerns/Questions and Points to Address in Rebuttal:

- Line 53 lacks evidence and most be supported through experiments or citations to prior work.

- How is this work different from [1] ?

- The concept of the Volatility Band (VB) appears intuitive—tokens near already decoded regions naturally exhibit higher confidence—so unless counterexamples or non-trivial cases are shown, the finding risks seeming self-evident rather than novel.


References:

[1] Wang, Xu, Chenkai Xu, Yijie Jin, Jiachun Jin, Hao Zhang, and Zhijie Deng. "Diffusion llms can do faster-than-ar inference via discrete diffusion forcing." arXiv preprint arXiv:2508.09192 (2025).

---

> ### Author Response · Authors · 2025-11-21
>
> We thank the reviewer for appreciating our motivation, elegant and lightweight method, and consistent accuracy improvement. Below, we address the questions raised by the reviewer.
>
> > Q1: Evaluation on text generation tasks and harder math reasoning tasks.
>
> We thank the reviewer for the suggestions. Following the suggestion, we evaluate AdaBlock-dLLM on two broader text generation tasks.
>
> For broader text generation tasks, we evaluate AdaBlock-dLLM on the IFEval (instruction following) [1] and Multi-News (summarization) [2] benchmarks using the delimiter set $\mathcal{D}$ ={`n`, `,`, `.`}. The results show that AdaBlock-dLLM yields performance gains, especially when block-level KV caching is applied.
>
> Table: Performance of AdaBlock-dLLM on IFEval (instruction-level loose accuracy) and Multi-News (ROUGE-L), using $B_0=32$.
>
> | IFEval |  |  |  |  |  |
> | :---- | :---- | :---- | :---- | :---- | :---- |
> | **Method** | **Vanilla (baseline)** | **Dynamic (Fast-dLLM)** | **\+Ada (our method)** | **\+Cache (Fast-dLLM)** | **\+Cache \+Ada (our method)** |
> | **Accuracy** | 66.67 | 66.67 | 67.51 (+0.84) | 64.63 | 66.43 (+1.8) |
> | **Multi-News** |  |  |  |  |  |
> | **Method** | **Vanilla (baseline)** | **Dynamic (Fast-dLLM)** | **\+Ada (our method)** | **\+Cache (Fast-dLLM)** | **\+Cache \+Ada (our method)** |
> | **ROUGE-L** | 27.33 | 27.31 | 27.51 (+0.2) | 26.41 | 26.95 (+0.54) |
>
> For more challenging math reasoning tasks such as AIME, we find that current diffusion language models like LLaDA-1.5 achieve zero or near-zero accuracy. We therefore leave the evaluation of stronger diffusion models to future work.
>
> > Q2: Reduced throughput at larger block sizes.
>
> We thank the reviewer for pointing this out. We attribute the slight overhead primarily to the change in effective block size.
>
> Our analysis suggests that using a larger block size can mitigate late-decoding overhead and thereby improve throughput. AdaBlock-dLLM attempts to align the block size with semantic steps, which typically leads to $B > B_0$ for smaller default block sizes $B_0 \in $ {4, 8} and $B < B_0$ for larger default block sizes $B_0 \in$ {16, 32, 64, 128}. This alignment between block boundaries and semantic units improves sampling quality and accuracy, while incurring only a minor throughput overhead relative to threshold-based dynamic sampling.
>
> We will include a more detailed discussion about the throughput in the next revision.
>
> > Q3: A more automatic way to select the delimiter threshold and delimiter set.
>
> We thank the reviewer for the suggestion. The delimiter threshold is tuned in a lightweight manner, and we additionally provide an automatic approach for selecting the delimiter set.
>
> For the delimiter threshold $ \tau_\mathcal{D} $, it is tuned only once on a single task and then reused across different tasks. Specifically, we calibrate its hyperparameter for each model using 100 samples from the GSM8K dataset and then apply the same hyperparameter across all benchmarks. This calibration step incurs only minor overhead and is amortized over all evaluations. We attribute the difference in the selected $ \tau_\mathcal{D} $ values to the discrepancy between adapted models (Dream) and models trained from scratch (LLaDA): the former exhibit more varied confidence distributions within the volatility band and therefore require a higher threshold to reliably indicate semantic boundaries.
>
> For the delimiter set, we propose an automated approach based on the occurrence of large confidence drops between consecutive tokens, denoted as $ \Delta c $. Using 100 samples from GSM8K, we observe that the token `\n` has the highest frequency of large confidence drops ($ \Delta c > 0.2 $) and predominantly leads the list. These large confidence drops indicate sharp semantic boundaries within the volatility band and are consistent with our motivation to align block sizes with semantic steps.
>
> We will include these details in the next version of the paper for clarity.
>
> Table: Top-10 tokens ranked by frequency of large confidence drops on GSM8K. A visualization is provided in Appendix A.4.
>
> | Token | "\\n" | " the" | "." | "," | "of" |
> | :---- | :---- | :---- | :---- | :---- | :---- |
> | **Frequency** | 1373 | 457 | 401 | 344 | 285 |
> | **Token** | **\[SPACE\]** | **"0"** | **"2"** | **":"** | **"5"** |
> | **Frequency** | 266 | 249 | 188 | 171 | 143 |
>
> [1] J. Zhou et al., “Instruction-Following evaluation for large language models,” arXiv.org, Nov. 14, 2023.
>
> [2] A. R. Fabbri, I. Li, T. She, S. Li, and D. R. Radev, “Multi-News: a Large-Scale Multi-Document Summarization Dataset and Abstractive Hierarchical Model,” arXiv.org, Jun. 04, 2019.

---

> > ### Author Response · Authors · 2025-11-21
> >
> > > Q4: Lack of discussion of situations when adaptive block size might fail.
> >
> > We thank the reviewer for the suggestions. We consider two cases:
> >
> > First, when the generation budget is small (e.g., in short MCQ-style tasks), semi-autoregressive decoding is not particularly beneficial. In such settings, AdaBlock-dLLM effectively reduces to standard decoding and is expected to yield similar performance.
> >
> > Second, AdaBlock-dLLM primarily improves the sampling quality. However, the decoding process of diffusion language models involves both denoising and sampling. If the denoiser is noisy or corrupted, the benefits of improved sampling are diminished, and the effectiveness of AdaBlock-dLLM correspondingly decreases.
> >
> > We will incorporate the discussion on limitations into the next version of the paper.
> >
> >
> > > Q5: Line 53 lacks evidence.
> >
> > We thank the reviewer for pointing this out. We provide experimental evidence for the premature decoding error.
> >
> > We quantified the sampling steps affected by premature decoding errors in Figure 5, which shows on GSM8K and HumanEval that around 10% of the decoding steps make suboptimal sampling decisions that lead to premature decoding errors. We will clarify this statement and include the appropriate reference in the next revision.
> >
> > > Q6: How is our work different from "Diffusion llms can do faster-than-ar inference via discrete diffusion forcing."
> >
> > The reviewer highlights the paper “Diffusion LLMs Can Do Faster-Than-AR Inference via Discrete Diffusion Forcing” (D2F), which improves decoding parallelism through an asymmetric distillation process. We clarify the difference between our method and D2F in terms of block scheduling and deployment efficiency:
> >
> > D2F allows a new block to participate in sampling before the previous block is fully decoded, effectively doubling the block size and allowing more tokens to be decoded in each step, increasing parallelism. In contrast, our method requires the previous block to be fully decoded, and then uses the confidence of delimiter tokens to predict the size of the next block. This enables adaptive block-size selection that aligns block boundaries with semantic steps.
> >
> > D2F increases parallelism by shaping the confidence distribution via additional training, so that more tokens exceed the sampling threshold in each denoising step. Our approach, by contrast, is purely inference-time: it optimizes the sampling schedule without any extra training and focuses on improving sampling quality through semantic-aware block scheduling. We view these two directions as complementary, and expect that AdaBlock-dLLM could be combined with D2F-style training in future work.
> >
> >
> > > Q7: The concept of Volatility Band (VB) lacks novelty.
> >
> > We thank the reviewer for pointing this out. Beyond the observation that tokens near the decoded region exhibit higher confidence (confidence locality), we further investigate the volatility band (VB) and identify semantically related local stochasticity, which motivates our block scheduler.
> >
> > Our analysis of confidence dynamics reveals a disaggregated global and local confidence pattern: although the decoding process of diffusion language models exhibits a global autoregressive tendency, the local decoding order is strongly driven by the semantic context and thus shows pronounced local stochasticity. Based on these observations, we identify the bottleneck induced by fixed block sizes in prevailing semi-autoregressive approaches and propose an adaptive block scheduler that improves sampling quality.
> >
> > We will revise the wording in the next version to better emphasize our main observations and clarify how they motivate the proposed methodology.

---

> > > ### Comment · Reviewer_caBK · 2025-11-23
> > > **Acknowledgement of Rebuttal**
> > >
> > > I thank the authors for the comprehensive experiments conducted during the rebuttal. The experiments address my questions well and I shall retain my score.

---

> > > > ### Author Response · Authors · 2025-11-24
> > > >
> > > > Dear Reviewer caBK,
> > > >
> > > > Thank you very much for your positive feedback and for carefully considering our rebuttal. We appreciate your time and are glad that the additional experiments satisfactorily addressed your concerns.

---

### Official Review · Reviewer_e6SK · 2025-10-31

**Soundness:** 3
**Presentation:** 3
**Contribution:** 2
**Rating:** 4
**Confidence:** 5

**Summary:**

This paper introduces a method to adaptively select the left and right boundaries for the block in the semi-AR setting for dLLM. The main idea come from the analysis of the confidence dynamics, where inside a volatility band area, the confidence fluctuates dynamically, and the VB regions exhibit semantics structure. Then the author propose to collect indices whose predicted tokens fall in the delimiter set to determine the block size. Experiments show that the performance exceeds the one in DualCache.

**Strengths:**

1. The paper proposes a new method to solve the problem of how to adaptively decide the block size and also its position in dLLMs.
2. The paper is well-written with sufficient analysis for the motivation.
3. The proposed method is effective compared to the baseline.

**Weaknesses:**

1. Adaptive block size is important, but the authors didn't show any other baseline for different ways to adaptively deciding the block size. They are other ways that can also achieve the adaptive block size, like some naive and straightforward ways you can just expanding the block size by using a sliding window and track the latest position of tokens that are not decoded as the left index. I think adding more analysis and comparisons with different ways to achieve adaptive block is needed.
2. Why the performance for MBPP on Dream-Base is only 12. The results from the official report is ~55%. It's a large discrepancy.
3. I am confusing about the result in Table 2. In Table 2, the TPS first increases and then decreases as you choose a larger bock size. In the analysis, the authors mentioned that it may because the increase of NFE, but the thing matters is the ratio between NFE and Block size. Can you provide more analysis on this.
4. I think the contribution of this submission is not that meet the standard of ICLR. I fully agree that adaptive block is needed, but the contribution of the method is only to choose the block size by the delimiter set, and they are a lot of papers discussing the confidence dynamics in dllm decoding. The idea is interesting, but it's too simple and straightforward.

**Questions:**

1. See Weaknesses.
2. The way of selecting the delimiter set can be explored more. For traditional NLP tasks, there are a lot of ways (semantic parsing) to know different types to separate the semantics of a sentence, not only by the rough way of selecting \n, [,], and [.].

---

> ### Author Response · Authors · 2025-11-21
>
> We thank the reviewer for appreciating our novel approach, detailed analysis, and effective method. We address the weaknesses and questions raised by the reviewer as follows.
>
> > Q1: Analysis and comparison with other adaptive block-size algorithms.
>
> We thank the reviewer for the suggestion. Below we summarize two representative variants that we implemented. It demonstrates that our method outperforms other adaptive variants.
>
> **Variant 1: Sliding Window Algorithm**
>
> In this approach, generation starts with a block of size $B$. Once a proportion $\alpha$ of tokens in the current block has been decoded, a new block of size $B$ is scheduled, starting from the first masked token. This is conceptually similar to the mechanism suggested by the reviewer.
>
> In the experiments, we observed little or no improvement in accuracy. In addition, this schedule risks poor integration with block-level KV caching: shifting block boundaries requires frequent KV cache eviction and re-computation, undermining the efficiency benefits of block-wise caching.
>
> Table: Adaptive block scheduling with sliding window.
>
> | Method | $B=16$ | $B=32$ | $B=64$ |
> | :---- | :---- | :---- | :---- |
> | **Dynamic** | 79.1 | 77.6 | 77.3 |
> | **\+Ada** | **80.6** | **80.6** | **79.5** |
> | **$\\alpha=0.25$** | 77.3 | 77.3 | 77.3 |
> | **$\\alpha=0.5$** | 77.0 | 77.0 | 77.0 |
> | **$\\alpha=0.75$** | 77.1 | 77.1 | 77.1 |
>
> **Variant 2: Shifting Schedule**
>
> In this approach, we apply a shifting schedule to allow imbalanced block sizes while keeping the total number of blocks fixed. For a given number of blocks $N$, generation length $L$, and shifting factor $\alpha$, we define
>
> $$
> f(t) = \frac{\alpha t}{1 + (\alpha - 1)t}, \qquad t \in [0, 1].
> $$
>
> The cumulative generated length at step $n \in \textbraceleft 1, \dots, N \textbraceright$ is
>
> $$
> L_n = L \times f\left(\frac{n}{N}\right), \qquad L_0 = 0,
> $$
> and the corresponding block size is $B_n = L_n - L_{n-1}$. Here, $\alpha > 1$ yields larger block sizes for smaller block indices (a front-loaded schedule), $\alpha = 1$ recovers a linear schedule, and $\alpha < 1$  yields smaller block sizes for larger block indices.
>
> Across a broad range of shifting factors, $\alpha \in \textbraceleft 0.1, 0.3, 0.5, 0.7, 3, 5 \textbraceright$, this scheme also did not yield measurable improvements in accuracy.
>
> Table: Adaptive block scheduling with shifting schedule.
>
> | Method | Acc. (%) | Method | Acc. (%) |
> | :---- | :---- | :---- | :---- |
> | **Dynamic** | 77.6 | **$\\alpha=0.5$** | 77.4 |
> | **\+Ada** | **80.6** | **$\\alpha=0.7$** | 78.0 |
> | **$\\alpha=0.1$** | 78.0 | **$\\alpha=3$** | 76.7 |
> | **$\\alpha=0.3$** | 77.0 | **$\\alpha=5$** | 74.4 |
>
> These negative results motivated us to systematically analyze the confidence dynamics and led to our proposed semantic-aware block scheduling strategies.
>
> > Q2: The performance discrepancy for MBPP on Dream-Base.
>
> For both LLaDA and Dream, we use the same postprocessing code for HumanEval and MBPP (from Fast-dLLM [1]) to ensure fair comparison.
>
> We further analyzed the raw generations and identified the primary cause of the discrepancy. The Dream-Base model has a particular tendency to produce multiple programs and frequently fails to output a correct `<EOS>` token. However, AdaBlock-dLLM still achieves better accuracy compared with the baselines.
>
> [1] C. Wu et al., “Fast-DLLM: Training-free acceleration of diffusion LLM by enabling KV cache and parallel decoding,” arXiv.org, May 28, 2025.

---

> > ### Author Response · Authors · 2025-11-21
> >
> > > Q3: The relationship between throughput, NFE, and Block size.
> >
> > We thank the reviewer for pointing this out. Below we provide an in-depth analysis of throughput, NFE, and block size.
> >
> > **NFE vs. Throughput.**
> > We find an approximately inverse relationship between NFE and throughput, both analytically and empirically. For diffusion language models such as LLaDA, NFE (number of function evaluations) is equal to the number of denoising operations. Since denoising (forward passes) dominates the computation cost, reducing the NFE directly reduces computation and thus improves throughput. Our experimental results, as shown in the table, also support this observation.
> >
> > **Throughput vs. Block Size.**
> > As observed in the table, for threshold-based dynamic sampling, throughput increases as the block size grows from 4 to 64, but then drops when the block size is further increased to 128. The throughput gains from 4 to 64 primarily come from mitigating the late decoding overhead. In contrast, the throughput drop from 64 to 128 is caused by noisy denoising outputs, aligning with the observed drop in accuracy. When the denoised sequence is noisy, fewer tokens achieve high confidence, requiring more denoising–sampling iterations and thus larger NFE. This NFE-induced overhead offsets the benefits of reduced late decoding overhead.
> >
> > **Throughput Analysis for AdaBlock-dLLM.**
> > The throughput of the “+Ada” setting is higher than that of “Dynamic” for small block sizes ($B_0 \in \textbraceleft 4, 8 \textbraceright$), and slightly lower for larger block sizes ($B_0 \in \textbraceleft 16, 32, 64, 128 \textbraceright$). This is consistent with the design of AdaBlock-dLLM, which tends to choose $B > B_0$ for small default block sizes and $B < B_0$ for larger default block sizes, so that block boundaries align with semantic steps. The corresponding improvements and slowdowns in throughput align with the above analysis of the interaction between block size, NFE, and throughput.
> >
> > We added more detailed results for Table 2 in the next version and provided an in-depth analysis.
> >
> > Table: Comparison of accuracy, throughput, average NFE, and the product of throughput and NFE across different methods.
> >
> > |  | Acc. (%) | TPS | Avg. NFE | TPS×NFE ($10^3$) | Acc. (%) | TPS | Avg. NFE | TPS×NFE ($10^3$) | Acc. (%) | TPS | Avg. NFE | TPS×NFE ($10^3$) |
> > | :---- | :---- | :---- | :---- | :---- | :---- | :---- | :---- | :---- | :---- | :---- | :---- | :---- |
> > | **$B\_0$** | **4** |  |  |  | **8** |  |  |  | **16** |  |  |  |
> > | **Vanilla** | 80.9 | 16.1 | 512.0 | 8.2 | 80.5 | 16.1 | 512.0 | 8.2 | 78.8 | 16.1 | 512.0 | 8.2 |
> > | **Dynamic** | 81.2 | 43.0 | 189.4 | 8.2 | 80.6 | 60.0 | 135.5 | 8.1 | 79.1 | 74.7 | 109.2 | 8.2 |
> > | **Dynamic+Ada** | 81.6 | 51.3 | 159.8 | 8.2 | 81.8 | 63.9 | 128.3 | 8.2 | 80.6 | 73.9 | 102.4 | 8.2 |
> > | **$B\_0$** | **32** |  |  |  | **64** |  |  |  | **128** |  |  |  |
> > | **Vanilla** | 76.8 | 16.1 | 512.0 | 8.2 | 76.8 | 16.1 | 512.0 | 8.2 | 71.0 | 16.0 | 512.0 | 8.2 |
> > | **Dynamic** | 77.6 | 85.0 | 94.9 | 8.1 | 77.3 | 89.4 | 91.6 | 8.2 | 70.7 | 81.2 | 101.2 | 8.2 |
> > | **Dynamic+Ada** | 80.6 | 83.5 | 98.5 | 8.2 | 79.5 | 87.9 | 93.4 | 8.2 | 72.8 | 80.6 | 101.6 | 8.2 |

---

> > > ### Author Response · Authors · 2025-11-21
> > >
> > > > Q4: Further explanation on our methodology and delimiter set selection
> > >
> > > We provide further clarification of our methodology and method for delimiter set selection below.
> > >
> > > AdaBlock-dLLM is derived from an in-depth systematic analysis of confidence dynamics in diffusion language models. We identify the high confidence plateau, the volatility band (VB), and the low confidence floor in the confidence pattern, and extend the analysis of the volatility band (VB). We identify local stochasticity that departs from the global autoregressiveness tendency and relate this phenomenon to the underlying semantic context.
> > >
> > > Our choice of delimiter set is directly supported by these insightful findings of confidence dynamics. We observe that delimiters such as `\n`, `,`, and `.` effectively track semantic boundaries, which manifest as pronounced confidence drops between consecutive positions, measured by $ \Delta c $. As shown in the table below, these delimiters have among the highest frequencies of large confidence drops (e.g., $ \Delta c > 0.2 $), indicating that the confidence values at these tokens are reliable indicators of semantic boundaries and thus well-suited for adaptive block scheduling.
> > >
> > > Table: Top-10 tokens ranked by frequency of large confidence drops on GSM8K. A visualization is provided in Appendix A.4.
> > >
> > > | Token | "\\n" | " the" | "." | "," | "of" |
> > > | :---- | :---- | :---- | :---- | :---- | :---- |
> > > | **Frequency** | 1373 | 457 | 401 | 344 | 285 |
> > > | **Token** | **\[SPACE\]** | **"0"** | **"2"** | **":"** | **"5"** |
> > > | **Frequency** | 266 | 249 | 188 | 171 | 143 |
> > >
> > > Although the final form of our method appears simple, it is **grounded in in-depth bottleneck identification** (late decoding overhead and premature decoding error) and **a systematic study of confidence dynamics**. We believe that identifying an approach that is both effective and low-cost at inference time is central to addressing the efficiency bottlenecks of diffusion language models. In particular, AdaBlock-dLLM is **the first adaptive block size scheduling technique** that preserves the efficiency benefits of both confidence-based dynamic sampling and block-level KV caching, without requiring additional training or modifications to the underlying model.
> > >
> > > During our exploration, we considered multiple alternative complicated strategies, including those mentioned in Q1, as well as approaches based on multiple parallel denoising trajectories to guide sampling. However, none of these alternatives outperforms the accuracy improvement and efficiency introduced by AdaBlock-dLLM.

---

> ### Author Response · Authors · 2025-11-27
> **Thank you and looking forward to further discussion**
>
> Dear Reviewer,
>
> Thank you again for your helpful feedback and constructive suggestions. We have done our best to address your comments with additional experiments and clarifications, and we hope our responses have resolved your main concerns.
>
> As the discussion period is coming to an end in less than a week, we would be grateful for any further comments or acknowledgment from you. Please let us know if any questions remain; we would be happy to provide additional clarification.

---

### Official Review · Reviewer_c59M · 2025-11-01

**Soundness:** 3
**Presentation:** 3
**Contribution:** 3
**Rating:** 4
**Confidence:** 2

**Summary:**

This paper identifies that fixed block sizes in semi-autoregressive (semi-AR) dLLM decoding cause "late decoding overhead" and "premature decoding error". It proposes AdaBlock-dLLM, a training-free scheduler that adaptively aligns block boundaries with semantic steps based on delimiter token confidence. This method improves accuracy by up to 5.3% without sacrificing throughput.

**Strengths:**

- The paper systematically analyze and address the limitations of the fixed block size assumption in semi-AR decoding.

- The core problems (late overhead, premature error) are clearly identified and illustrated. The proposed solution is simple, intuitive, and well-supported by experiments.

- The method is practical (training-free, plug-and-play) and shows clear accuracy improvements on the accuracy-throughput frontier, especially when combined with KV caching.

**Weaknesses:**

- The method introduces new hyperparameters that require model-specific tuning, slightly weakening the "plug-and-play" claim.

**Questions:**

Given the marginal benefit of adding more delimiters (Table 6), did you consider a more automated or learned method to identify semantic boundaries instead of a predefined list?

---

> ### Author Response · Authors · 2025-11-21
>
> We thank the reviewer for appreciating our problem identification and clear improvement. Below, we address the questions raised by the reviewer.
>
> > Q1: Model-specific tuning slightly weakens the “plug-and-play” claim.
>
> We thank the reviewer for pointing out. The delimiter threshold $ \tau_\mathcal{D} $ **is tuned only once on a single task** and then reused across different tasks, indicating that our approach is relatively insensitive to hyperparameter tuning.
>
> Specifically, we select $ \tau_\mathcal{D} $ using a 100-sample calibration subset of the GSM8k dataset and then reuse this value for the same model across all benchmarks. This calibration step is lightweight, performed only once per model in one task, and then reused across different benchmarks, where its cost is amortized over all subsequent evaluations.
>
> The model-specific threshold primarily accounts for differences between adapted models (e.g., Dream) and models trained from scratch (e.g., LLaDA), with the former exhibiting substantially higher confidence variance. We will clarify this calibration procedure and its motivation in the next version of the paper.
>
> > Q2: Consider an automated or learned method to identify the semantic boundaries.
>
> We thank the reviewer for the suggestion. We provide **an automated approach** to identify the semantic boundaries.
>
> An automated approach can be designed based on the occurrence of large confidence drops between consecutive tokens, denoted as $ \Delta c $. Using 100 samples from the GSM8K dataset, we observe that the tokens `\n` have the highest frequency of large confidence drops ($ \Delta c > 0.2 $) and predominantly lead the list. This behavior indicates sharp semantic boundaries at these positions and supports our design choice of using them as delimiters to align block segmentation with semantic steps.
>
> Table: Top-10 tokens ranked by frequency of large confidence drops on GSM8K. A visualization is provided in Appendix A.4.
>
> | Token | "\\n" | " the" | "." | "," | "of" |
> | :---- | :---- | :---- | :---- | :---- | :---- |
> | **Frequency** | 1373 | 457 | 401 | 344 | 285 |
> | **Token** | **\[SPACE\]** | **"0"** | **"2"** | **":"** | **"5"** |
> | **Frequency** | 266 | 249 | 188 | 171 | 143 |
>
> We agree that learned methods for discovering semantic boundaries are a promising direction and will explore them in future work.

---

> ### Author Response · Authors · 2025-11-27
> **Thank you and looking forward to further discussion**
>
> Dear Reviewer,
>
> Thank you again for your helpful feedback and constructive suggestions. We have done our best to address your comments with additional experiments and clarifications, and we hope our responses have resolved your main concerns.
>
> As the discussion period is coming to an end in less than a week, we would be grateful for any further comments or acknowledgment from you. Please let us know if any questions remain; we would be happy to provide additional clarification.

---

### Official Review · Reviewer_THAi · 2025-11-02

**Soundness:** 2
**Presentation:** 2
**Contribution:** 2
**Rating:** 6
**Confidence:** 3

**Summary:**

This paper presents AdaBlock-dLLM, a semantic-aware adaptive block-size decoding method for diffusion-based large language models (dLLMs). By analyzing confidence dynamics during denoising, the authors propose AdaBlock-dLLM, which is a training-free, plug-and-play scheduler that dynamically adjusts block boundaries according to semantic step length. Extensive experiments on benchmarks show that the proposed approach maintains compatibility with caching mechanisms and improves both efficiency and semantic consistency in diffusion LLM inference.

**Strengths:**

1. This paper proposes an adaptive, training-free approach that integrates seamlessly with existing diffusion LLM frameworks and improves accuracy without retraining.

2. This paper provides a clear and well-motivated analysis of confidence dynamics and their connection to semantic structures in dLLM decoding, then uses the confidence score of delimiter to decide the block size adaptively.

**Weaknesses:**

1. The heuristic nature of the delimiter-based semantic segmentation might not generalize well to less structured text other than math or coding problems. The authors should evaluate on more diverse test sets. For example, what if the output text is expected to be a long section and there's no '\n' in the output?

2.  Figure 6 can hardly be viewed as a pareto-frontier, as the throughput nearly does not change as the accuracy is increasing.

**Questions:**

Please refer to weakness for more details.

---

> ### Author Response · Authors · 2025-11-21
>
> We thank the reviewer for appreciating our seamless integration and well-motivated analysis. We address the reviewer’s questions as follows.
>
> > Q1: AdaBlock-dLLM does not generalize well to tasks other than math or coding benchmarks.
>
> We thank the reviewer for the suggestion. Following this suggestion, we evaluate AdaBlock-dLLM on the IFEval (instruction following) [1] and Multi-News (summarization) [2] benchmarks using the delimiter set $\mathcal{D}$ \={`\n`, `,`, `.`}. The results indicate that AdaBlock-dLLM generalizes beyond math and coding benchmarks and provides more pronounced benefits when combined with block-level KV caching.
>
> For reasoning tasks, the selection of `\n` aligns well with the CoT-style prompting, which typically structures each reasoning step on a separate line. AdaBlock-dLLM also supports extending the delimiter set to include other tokens (e.g., periods and commas). These additional delimiters help align block sizes with semantic steps and thereby improve sampling quality beyond strictly structured reasoning tasks.
>
> Table: Performance of AdaBlock-dLLM on IFEval (instruction-level loose accuracy) and Multi-News (ROUGE-L), using $B_0=32$.
> | IFEval |  |  |  |  |  |
> | :---- | :---- | :---- | :---- | :---- | :---- |
> | **Method** | **Vanilla (baseline)** | **Dynamic (Fast-dLLM)** | **\+Ada (our method)** | **\+Cache (Fast-dLLM)** | **\+Cache \+Ada (our method)** |
> | **Accuracy** | 66.67 | 66.67 | 67.51 (+0.84) | 64.63 | 66.43 (+1.8) |
> | **Multi-News** |  |  |  |  |  |
> | **Method** | **Vanilla (baseline)** | **Dynamic (Fast-dLLM)** | **\+Ada (our method)** | **\+Cache (Fast-dLLM)** | **\+Cache \+Ada (our method)** |
> | **ROUGE-L** | 27.33 | 27.31 | 27.51 (+0.2) | 26.41 | 26.95 (+0.54) |
>
>
> > Q2: The Pareto-frontier notation in Figure 6
>
> We thank the reviewer for this suggestion.
> - On the GSM8K and MATH datasets, AdaBlock is the best configuration, reaching 80.7% and 37.4% accuracy at throughputs of 160.6 and 68.7 tps, respectively.
> - On the MBPP dataset, AdaBlock accounts for 4 out of 6 of the points on the Pareto frontier.
> - On the HumanEval dataset, two AdaBlock configurations are still on the Pareto frontier in the high-accuracy regime, which we consider the more practical settings. In the low-accuracy regime, throughput differences are small and effectively negligible.
>
> In response to the reviewer’s comment, we will refine the Pareto-frontier notation and revise the corresponding phrasing for HumanEval and MBPP in the next version of the paper.
>
> [1] J. Zhou et al., “Instruction-Following evaluation for large language models,” arXiv.org, Nov. 14, 2023.
>
> [2] A. R. Fabbri, I. Li, T. She, S. Li, and D. R. Radev, “Multi-News: a Large-Scale Multi-Document Summarization Dataset and Abstractive Hierarchical Model,” arXiv.org, Jun. 04, 2019.

---

### Author Response · Authors · 2025-11-21

We sincerely thank the reviewer for their insightful suggestions. In the revised version, we highlight the modified part in **Orange** to make it easier to identify the changes.

We would be grateful for any acknowledgement or discussion from the reviewers. Please let us know if any questions remain, as we would be happy to provide further clarification.

---

### Comment · Area_Chair_kpRA · 2025-11-22
**Action Needed: Review Rebuttal and Update Evaluation**

Dear Reviewers,

Thank you, as always, for your valuable contributions and efforts. The authors have now submitted their rebuttal. Please take a moment to review it and provide any necessary follow-up actions, such as additional questions, clarification requests, or updates to your review.

Since the initial ratings ranged from 4 to 8, I kindly ask you to pay close attention to the perspectives of the other reviewers when preparing your final response.

Thank you again for your support.

---

### Author Response · Authors · 2025-12-03
**Summary of Discussion Period**

Dear Area Chair,

We sincerely appreciate the time and effort you have devoted to overseeing the review process for our submission. During the rebuttal stage, the reviewers acknowledged the clear motivation, the effectiveness of our training-free method, and the good quality of the presentation. Below, we summarize the key points that emerged during the discussion period.

---

## Summary of our work

Our work targets improving the sampling accuracy of diffusion-based LLM (dLLM) inference. We revisit the conventional semi-autoregressive (semi-AR) decoding paradigm with fixed block sizes and identify two fundamental bottlenecks: **Late-decoding Overhead** and **Premature-decoding Error**, which motivate the need for adaptive block sizing. To design the adaptive scheduler, we systematically analyze confidence dynamics during decoding and uncover the **volatility band** (VB), a stochastic region strongly governed by local semantic context. Leveraging this insight, we introduce an adaptive block-size scheduler (*AdaBlock-dLLM*) that aligns block boundaries with semantic boundaries via a set of delimiter tokens, thereby reducing both overhead and errors. Across extensive experiments, our method improves accuracy by **up to 5.3%** on multiple tasks, especially when combined with blockwise KV caching, while maintaining throughput comparable to state-of-the-art baselines. To our knowledge, this is **the first work** to challenge the fixed block size assumption for dLLMs, and our method is fully compatible with existing techniques for improving dLLM inference efficiency.

---

## Strengths of our work acknowledged by reviewers
Reviewers have recognized our work as a simple yet effective improvement to the current semi-autoregressive decoding paradigm, specifically highlighting:

- **Clear motivation.** Clearly identifies late-overhead, premature-error issues and fixed block-size inefficiencies, and uses confidence dynamics/semantic structure to motivate delimiter-based adaptive scheduling. (Reviewers: THAi, c59M, e6SK, caBK)

- **Adaptive, training-free solution.** Introduces a training-free, delimiter-driven block-size scheduler that is plug-and-play with existing diffusion LLM and semi-AR frameworks. (Reviewers: THAi, c59M, e6SK, caBK)

- **Effective and practical performance.** Lightweight method that consistently improves accuracy and accuracy–throughput trade-offs, especially with KV caching, across multiple models and datasets. (Reviewers: THAi, c59M, e6SK, caBK)

- **Good writing and presentation.** Method and motivation are clearly presented with a conceptually elegant description of the adaptive scheduler’s role in diffusion LLMs. (Reviewers: e6SK, caBK)

(to be continued)

---

> ### Author Response · Authors · 2025-12-03
> **Summary of Discussion Period (continued)**
>
> (continued)
>
> ## Common concerns from the reviewers and our clarifications
>
> We summarize three common concerns raised by the reviewers and provide our clarifications below.
>
> > **Generalization to more datasets. (Reviewers: THAi, caBK)**
>
> We additionally evaluate AdaBlock-dLLM on two broader text generation benchmarks: IFEval (instruction following) and Multi-News (summarization). In both cases our method consistently **improves performance over the baselines**, supporting the generality of our approach.
>
> | IFEval |  |  |  |  |  |
> | :---- | :---- | :---- | :---- | :---- | :---- |
> | **Method** | **Vanilla (baseline)** | **Dynamic (Fast-dLLM)** | **\+Ada (our method)** | **\+Cache (Fast-dLLM)** | **\+Cache \+Ada (our method)** |
> | **Accuracy** | 66.67 | 66.67 | 67.51 (+0.84) | 64.63 | 66.43 (+1.8) |
> | **Multi-News** |  |  |  |  |  |
> | **Method** | **Vanilla (baseline)** | **Dynamic (Fast-dLLM)** | **\+Ada (our method)** | **\+Cache (Fast-dLLM)** | **\+Cache \+Ada (our method)** |
> | **ROUGE-L** | 27.33 | 27.31 | 27.51 (+0.2) | 26.41 | 26.95 (+0.54) |
>
> > **Analysis on throughput performance (Reviewers: e6SK, caBK)**
>
> We provide additional analysis of throughput performance, with detailed explanations in Section 5.2 of the revised version. Here we summarize three main insights:
>
> - Throughput exhibits an inverse relationship with the number of function evaluations (NFE), since NFE corresponds to the number of forward passes, which dominate the computational cost.
>
> - As the block size $B$ increases, throughput improves due to reduced late-decoding overhead. However, when $B$ becomes overly large (e.g., $B = 128$), semantic coherence weakens, which increases NFE and consequently reduces throughput; this effect is also reflected in significantly lower accuracy at overly large $B$.
>
> - AdaBlock-dLLM tends to increase $B$ when the the initial block size is small ($B_0 \in$ {4, 8}), thereby improving throughput. For larger initial block sizes ($B_0 \in$ {16, 32, 64, 128}), the selected $B$ is typically smaller than $B_0$, resulting in a slight throughput decrease, while the overhead remains minor.
>
> > **Selection of hyperparameters (delimiter set $\mathcal{D}$, delimiter threshold $\tau_{\mathcal{D}}$) (Reviewers: c59M, e6SK, caBK)**
>
> For the selection of the delimiter set $\mathcal{D}$, we provide **an automated procedure** that supports the delimiter choices used in our experiments. By measuring the occurrence of large confidence drops between consecutive tokens, denoted as $\Delta c$, we observe that the token `\n` has the highest frequency of large confidence drops ($\Delta c > 0.2$) and predominantly leads the list (Figure 9 in the revised version). This behavior indicates sharp semantic boundaries at these positions and supports our design choice of using such tokens as delimiters to align block segmentation with semantic steps.
>
> The hyperparameter $\tau_{\mathcal{D}}$ is tuned **only once on a single task** and then reused across different tasks. We use a 100-sample calibration subset of the GSM8K dataset to determine $\tau_{\mathcal{D}}$, and the overhead of this lightweight calibration step is amortized over all subsequent evaluations. Additionally, in Section 5.3, we justify using different $\tau_{\mathcal{D}}$ values for different model families by attributing this difference to the distinct confidence distributions.
>
> ---
>
> We hope that this summary of our previous discussions provides the necessary context for your decision-making regarding our paper. We thank you again for your time and additional efforts throughout the review process.
>
> Best regards,
>
> The Authors

---

### Meta-Review · Area_Chair_hTCf · 2026-01-04

**Summary:**

Reviewer THAi expressed concerns about the generality of the proposed method, noting that the delimiter-based semantic segmentation may not transfer well beyond structured domains such as math or coding, especially in cases where the generated text is long and lacks clear delimiters like line breaks. This reviewer also questioned the interpretation of the Pareto frontier results, pointing out that in some figures the throughput changes were minimal despite accuracy improvements.

Reviewer c59M acknowledged the clarity of the problem formulation and the practicality of the approach, but raised concerns that the introduction of model-specific hyperparameters weakens the plug-and-play claim. This reviewer also questioned whether relying on a predefined delimiter list is optimal, and suggested exploring more automated or learned methods for identifying semantic boundaries.

Reviewer e6SK was more critical of the contribution, arguing that while adaptive block sizing is important, the proposed method appears overly simple and lacks comparisons with other plausible adaptive block-size strategies. This reviewer questioned the novelty of the approach, highlighted unexplained discrepancies in benchmark results, and requested deeper analysis of the relationship between block size, NFE, and throughput, as well as a more principled discussion of delimiter selection.

Reviewer caBK was generally positive but noted several limitations, including the narrow focus on math and code generation tasks, reduced throughput at larger block sizes, and the manual tuning of delimiter thresholds and sets. This reviewer also asked for clearer differentiation from closely related prior work and for stronger justification of the novelty of the Volatility Band concept, as well as discussion of scenarios where adaptive block sizing might fail.

**Reviewer Concerns:**

This meta-reviewer believes that several of the most important and commonly raised concerns were substantially addressed in the rebuttal. In particular, the concern regarding limited evaluation beyond math and code benchmarks was directly handled through additional experiments on instruction-following and summarization tasks, which demonstrated that the proposed method generalizes to broader text generation settings, especially when combined with block-level KV caching. The rebuttal also convincingly addressed questions about the heuristic nature of delimiter selection by providing a data-driven analysis based on confidence drops, as well as an automated criterion for identifying semantic boundaries. In addition, concerns about the lack of comparison to other adaptive block-size strategies were meaningfully addressed through the implementation and evaluation of multiple alternative adaptive scheduling variants, which did not yield comparable improvements and helped clarify the necessity of the proposed design.

This meta-reviewer further finds that the rebuttal provided a clear and technically grounded explanation of the interaction between block size, NFE, and throughput, resolving earlier confusion about throughput trends at larger block sizes. The discussion of failure modes and limitations, including cases with short generation budgets or noisy denoisers, also addressed requests for a more balanced and realistic assessment of the method’s applicability. Clarifications regarding benchmark discrepancies, such as the MBPP results on Dream-based models, were supported by concrete analysis of generation behaviors and post-processing effects, which helped remove ambiguity around these results.

Some concerns remain only partially addressed. While the rebuttal clarified how delimiter thresholds are calibrated and amortized across tasks, the approach still relies on model-specific calibration rather than a fully learned or parameter-free mechanism. Similarly, although the rebuttal sharpened the distinction between this work and prior diffusion-based decoding methods and elaborated on the non-trivial interpretation of the volatility band, the perceived simplicity of the core idea may still limit how strongly its novelty is viewed by some reviewers. Nevertheless, this meta-reviewer considers the remaining concerns to be relatively minor in nature and largely related to scope and positioning rather than to technical correctness or empirical validity.

**Reviewer Scores:**

All in all, this meta-reviewer believes that the rebuttal meaningfully addressed the core concerns that led the two reviewers to assign borderline scores. The most prominent issues raised in those reviews focused on the lack of comparisons with alternative adaptive block-size strategies, unclear throughput behavior at larger block sizes, and the perceived heuristic nature of delimiter selection.

The rebuttal responded to these points with concrete additions and clarifications. By implementing and evaluating multiple alternative adaptive scheduling variants, the authors showed that simpler or more naive approaches either fail to improve accuracy or do not integrate well with block-level KV caching, thereby justifying the specific design choices of the proposed method. The detailed analysis of the relationship between block size, NFE, and throughput resolved earlier confusion about scalability and performance trade-offs. Furthermore, the clarification that delimiter thresholds are calibrated only once per model and reused across tasks, together with a data-driven justification for delimiter selection based on confidence drops, substantially reduced concerns about ad-hoc heuristics.

Overall, these responses strengthen confidence in the technical soundness and practical relevance of the approach, and suggest that the two reviewers who initially gave borderline scores would have been inclined to view the paper more favorably had they been able to fully participate in the discussion.

---

### Decision · Program_Chairs · 2026-01-26

Accept (Poster)